# Health-related quality of life of younger and older lower-income households in Malaysia

**Hussein Rizal**[1,2], **Mas Ayu Said**[1,3]*, **Hazreen Abdul Majid**[1,2,4], **Tin Tin Su**[5], **Tan Maw Pin**[6], **Rozmi Ismail**[7], **Mohd Azlan Shah Zaidi**[8]

**1** Department of Social and Preventive Medicine, Faculty of Medicine, University of Malaya, Kuala Lumpur, Malaysia, **2** Centre for Population Health (CePH), Department of Social and Preventive Medicine, Faculty of Medicine, University of Malaya, Kuala Lumpur, Malaysia, **3** Centre for Epidemiology and Evidence-Based Practice, Department of Social and Preventive Medicine, Faculty of Medicine, University of Malaya, Kuala Lumpur, Malaysia, **4** Department of Nutrition, Faculty of Public Health, Universitas Airlangga, Jawa Timur, Indonesia, **5** South East Asia Community Observatory (SEACO) and Global Public Health, Jeffrey Cheah School of Medicine & Health Sciences, Monash University, Bandar Sunway, Subang Jaya, Malaysia, **6** Department of Medicine, Faculty of Medicine, University of Malaya, Kuala Lumpur, Malaysia, **7** Faculty of Social Sciences and Humanity, National University of Malaysia, Bangi, Selangor, **8** Faculty of Economics and Management, National University of Malaysia, Bangi, Selangor

* mas@ummc.edu.my

**Data Availability Statement:** All relevant data are within the paper and its Supporting Information files.

## Abstract

### Background

Globally, a lower income is associated with poorer health status and reduced quality of life (QOL). However, more research is needed on how being older may influence QOL in lower-income households, particularly as older age is associated with an increased risk of chronic diseases and care needs. To this end, the current study attempts to determine the health-related QOL (HRQOL) among individuals from lower-income households aged 60 years and over compared to lower-income adults aged less than 60 years.

### Methods

Participants were identified from the Department of Statistics Malaysia sampling frame. Surveys were carried out with individual households aged 18 years and older through self-administered questionnaires. Information was collected on demographics, household income, employment status, number of diseases, and HRQOL assessed using the EuroQol 5-Dimension 5-Level (EQ-5D-5L) tool.

### Results

Out of a total of 1899 participants, 620 (32.6%) were female and 328 (17.3%) were aged 60 years and above. The mean (SD) age was 45.2 (14.1) and mean (SD) household income was RM2124 (1356). Compared with younger individuals, older respondents were more likely to experience difficulties in mobility (32.1% vs 9.7%, $p<0.001$), self-care (11.6% vs 3.8%, $p<0.001$), usual activities (24.5% vs 9.1%, $p<0.001$), pain/discomfort (38.8% vs 16.5%, $p<0.001$) and anxiety/depression (21.4% vs 13.5%, $p<0.001$). The mean (SD) EQ-5D index scores were lower among older respondents, 0.89 (0.16) vs 0.95 (0.13), $p = 0.001$.

**Funding:** This research was funded by the Ministry of Higher Education via the Long Term Research Grant (LRGS) awarded to the Malaysia Research University Network (MRUN) with the grant code of LRGS/1/2016/UKM/02/1/2 (LGRS MRUN/F1/01/2019). The funder played no role in the study design, data collection and analysis, decision to publish, or preparation of the manuscript.

**Competing interests:** The authors have declared that no competing interests exist.

After adjusting for covariates, age was a significant influencing factor ($p = 0.001$) for mobility (OR = 2.038, 95% CI:1.439–2.885), usual activities (OR = 1.957, 95% CI:1.353–2.832) and pain or discomfort (OR = 2.241, 95% CI:1.690–2.972).

## Conclusion

Lower-income older adults had poorer HRQOL compared to their younger counterparts. This has important implications concerning intervention strategies that incorporate active ageing concepts on an individual and policy-making level to enhance the QOL and wellbeing, particularly among the older lower-income population.

## Introduction

Several studies found that quality of life (QOL) in their study population worsens with increasing age [1, 2]. Older people have higher probability of suffering from multiple health disorders due to experiencing reduced physical and mental functions. Loneliness, impaired sexual activity and chronic metabolic disorders are some of the causes that can result in emotional disturbances which lead to reduced QOL [3]. This is of particular concern with global population ageing, as population ageing is accelerated in many middle-income developing nations, and most older persons now reside in lower and middle-income countries [4]. The World Health Organisation (WHO) defines QOL as "an individual's perception of their position in their life in the context of the culture and value systems in which they live and in relation to their goals, expectations, standards and concerns" [5]. It is a broad concept that affects the individual in a complicated way, including physical health, psychological status, social relationships, and their relationship with their surrounding environment.

Enhancing and maintaining the QOL as a person ages is the primary goal of active ageing [6]. The ageing well concept is viewed as a combination of good health status, participation in paid and unpaid activities, and assurance of social, financial, and physical security [7]. As the population becomes older, there is an increasing burden of chronic diseases, the risk of an unhealthy lifestyle and economic challenges. Thus, interest in tracking the changes in QOL has increased in the research community. The findings from studies that focus on these issues may provide evidence-based measures that can be drawn on to promote active ageing, which focuses on physical and mental health and a broader dimension that includes other contributing factors such as social interactions and the environment [8–10]. One effective instrument to measure health-related QOL (HRQOL) is the EuroQol 5-Dimension 5-Level (EQ-5D-5L) tool [11]. Aside from its robust validity and reliability, the tool is short and uses simple language, making it easy to answer and suitable for widescale use among Malaysians [11–14]. Studies of EQ-5D in Malaysia used non-Malaysian tariffs to obtain utility health states [12, 15], sampled among the general population [12, 13], patient setting [16, 17], and low-income groups [18].

Analysis of the EQ-5D responses by demographic variables found significant differences among categories of age, gender, and self-reported chronic medical conditions [19, 20]. Other studies using the EQ-5D also demonstrated lower scores in older individuals than younger ones [19, 21], lower scores in women than in men, and lower scores in individuals of lower economic status compared with higher socioeconomic status [22, 23]. As the presence of chronic diseases is strongly linked with physical health and QOL [24], the increasing trend of chronic diseases reported by the National Health Morbidity Survey (NHMS) is more detrimental to the lower-income group [25]. Previous studies comparing sociodemographic status and HRQOL consistently showed that low income is associated with low HRQOL despite

differing countries and samples, even after controlling for risk factors such as chronic diseases [26, 27].

Empirical analyses concerning HRQOL could further extend the understanding of health inequalities. They indicate that low health status groups are faced with a double burden, first by increased levels of health impairments and second by lower levels of HRQOL once health is impaired [26]. The interactions between differing kinds of inequality and its factors are often complex, multidirectional, and inter-related. People can find it more difficult to move away from unhealthy behaviours if they are worse off in a wider range of determinants of health. For example, lower socio-economic groups tend to have a higher prevalence of risky health behaviours, poor diet, limited access to care and less opportunity to lead healthy lives. In addition, disadvantages are concentrated in particular sectors of the population and can be mutually reinforcing [28]. This means that the low-income population living in close proximity together reinforces their accumulative disadvantage such as being in their psychological comfort zone, lower education quality in terms of access to better schools and to afford tuitions, and limited occupational opportunities due to lower education and skill level.

These issues have been exacerbated since the onset of the COVID-19 pandemic, which has worsened the situation due to the rapid decline in the economy [20]. Given that people with lower socioeconomic status were the most affected by rising non-communicable diseases (NCDs), the objective of this study is to determine the differences in HRQOL within lower-income households in Malaysia, particularly among younger and older individuals. This study also attempts to identify factors influencing HRQOL.

## Methods

### Study sample

The Department of Statistics Malaysia (DOSM) utilised a simple random sampling method to identify eligible participants aged 18 years and older. Low-income participants were identified as being in the Bottom 40 (B40) group characterised as having an income of less than RM4,850 [29]. Household income information was used to identify participants in the B40 income group. With a total population of 32.75 million as of the first quarter of 2021 [30], 2125 participants were recruited from six Malaysian states, including Selangor, Pahang, Sabah, Sarawak, Pulau Pinang and Johor. A total of six out of the 13 states were selected through the representation of each region [29]. For example, in the Western Peninsula, the central region is Selangor, the northern region is Pulau Pinang, the southern region is Johor, and the eastern region is Pahang. Whereas the Eastern Peninsula is Sabah and Sarawak. The municipalities of Seberang Perai, Selayang, Ampang, Subang Jaya, Shah Alam, Petaling Jaya, Johor Bahru, Kuantan, Kuching Utara, Kuching Selatan, Padawan dan Kota Kinabalu were randomly selected in each respective state (S1 Table).

### Study design and duration

A cross-sectional study was conducted from 1st September 2020 until 24th October 2020 during the COVID-19 pandemic. Due to an increase in cases, data collection was continued from January 25th until February 15th, 2021. Enumerators acquired signed consent and administered the self-reported questionnaire to the participants.

### Data collection

The questionnaire consisted of categorical questions such as gender, ethnicity, educational level, marital status, occupational status, total household income, number of diseases and

QOL. The reported chronic diseases cover eight major medical conditions such as hypertension, diabetes mellitus, cardiovascular disease, kidney failure, cancer, asthma, stroke, and mental illness.

## Quality of life instrument

The translated validated Malay version of the EuroQol 5-Dimension 5-Level (EQ-5D-5L) tool (S2 Table) consists of a descriptive system and visual analogue scale (EQ-VAS) [11]. The descriptive system classifies health into five distinct dimensions, which includes: (1) Mobility, (2) Self-care, (3) Usual activities, (4) Pain/discomfort and, (5) Anxiety/depression [31, 32]. Within each dimension, participants were asked to describe their perceived current health status using five levels (Likert scale) of severity comprising of: (1) Having no problems, (2) Having slight problems, (3) Having moderate problems, (4) Having severe problems and, (5) Unable to/having extreme problems [31, 32]. The EQ-VAS, on the other hand, is a hash-marked vertical scale ranging from 0 to 100, in which 0 represents the worst imaginable health (death) and 100 for the best imaginable health. It is a continuous scale that measures a person's subjective health. The EQ-5D-5L instrument was validated in the general population and the patient group setting and thus encouraged the application in health-related research in Malaysia [11–13]. The data collected can be presented in three ways: (1) as a health profile, (2) as a measure of overall self-rated health status and (3) as an index value. The Malaysian EQ-5D value set was applied to calculate values of all health states generated by the tool, which has a range of -0.442 to 1 [13, 14]. The value set for the original version of the tool ranged from 0 to 1.

## Data analysis

Frequency analysis was carried out to identify missing values and extreme values. Extreme values that were deemed as a case of wrong data entry were deleted and assigned a missing value. The main outcomes described in this study includes the EQ-5D-5L values and each of its dimensions and EQ-VAS. The EQ-5D-5L values represented the HRQOL and were presented descriptively and in the form of an index score. The continuous variable of age was dichotomised into two groups: younger, characterised as an individual aged 18 to 59 years old and older, characterised as an individual aged 60 years or older, respectively (Tables 1 and 3). Each dimension of the five-level model was dichotomised to "no problems at all", defined as a score of one and "some to severe problems" defined as a score of two. Normality test using Shapiro Wilk test showed that the EQ-5D-5L and EQ-VAS were not normally distributed ($p < 0.05$) and thus were tested using the Mann-Whitney U test and Kruskal Wallis test to identify the difference in HRQOL between the younger and older group. Further analysis of logistic regression was conducted for each dichotomised dimension of EQ-5D and is used as the dependent variable. The Chi-square test was used to measure the relationship of each dependent variable. Age category and number of diseases were identified as the primary indicators, followed by gender, ethnicity, marital status, educational status, and employment status. Statistical significance was set at a two-sided *p*-value of 0.05. The backward stepwise regression was used, which is a stepwise regression approach that begins with a full (saturated) model and at each step gradually eliminates variables from the regression model to find a reduced model that best explains the data. This method was applied to each dimension of EQ-5D. The cut-off value of 0.50 for the regression, odds ratio and 95% confidence interval were applied. All relevant data are within the manuscript and its supporting information file (S1 Appendix).

## Ethical considerations

A consent form was given and signed by the participants before answering the survey. The study was approved by the University of Malaya Research Ethics Committee (UMREC) (Reference number: UM.TNC2/UMREC_1215). This study prescribes the ethics upheld by the Helsinki Declaration.

## Results

### Characteristics of the participants

Out of the 2125 participants recruited, a total of 1899 completed the questionnaire. The mean and standard deviation (SD) for age was 45.2 (14.1) years. The frequency and percentage for males were 1279 (67.4%) and for females were 620 (32.6%); Malay and Bumiputera were 1564 (83.2%), Chinese were 156 (8.3%) and Indian were 159 (8.5%); married individuals were 1348 (71.1%), unmarried were 269 (14.2%) and divorced or separated were 279 (14.7%); currently employed were 1333 (70.2%) and currently unemployed were 566 (29.8%); individuals without any chronic disease were 1336 (70.4%), with one chronic disease were 357 (18.8%), with two chronic diseases were 165 (8.7%) and with three or more chronic diseases were 41 (2.1%); highest education at primary level were 578 (31.5%), secondary were 780 (42.6%), pre-university were 303 (16.5%) and university were 172 (9.4%). Table 1 further characterises participants into the younger (aged 18–59 years old) and older (aged 60 years and above) groups.

**Table 1. Characteristics of younger and older participants.**

| Description | Younger (18–59 years old) | | | Older (≥60 years old) | | | p-value |
|---|---|---|---|---|---|---|---|
| | n | % | Mean (SD) | n | % | Mean (SD) | |
| **Age (year)** | | | 40.76 (10.83) | | | 66.58 (5.85) | **0.001** |
| **Gender** | 1571 | 82.7 | | 328 | 17.3 | | 0.991 |
| Male | 1058 | 67.3 | | 221 | 67.4 | | |
| Female | 513 | 32.7 | | 107 | 32.6 | | |
| **Ethnicity** | 1556 | 82.8 | | 323 | 17.2 | | **0.001** |
| Malay / Bumiputera | 1326 | 85.2 | | 238 | 73.7 | | |
| Chinese | 102 | 6.6 | | 54 | 16.7 | | |
| Indian | 128 | 8.2 | | 31 | 9.6 | | |
| **Marital Status** | 1568 | 82.7 | | 328 | 17.3 | | **0.001** |
| Married | 1120 | 71.4 | | 228 | 69.5 | | |
| Unmarried | 261 | 16.6 | | 8 | 2.5 | | |
| Divorced/separated | 187 | 12.0 | | 92 | 28.0 | | |
| **Employment** | 1571 | 82.7 | | 328 | 17.3 | | **0.001** |
| Employed | 1238 | 78.8 | | 95 | 29.0 | | |
| Unemployed | 333 | 21.2 | | 233 | 71.0 | | |
| **Chronic Disease** | 1571 | 82.7 | | 328 | 17.3 | | **0.001** |
| 0 | 1190 | 75.7 | | 146 | 44.5 | | |
| 1 | 251 | 16.0 | | 106 | 32.3 | | |
| 2 | 106 | 6.7 | | 59 | 18.0 | | |
| 3 or more | 24 | 1.6 | | 17 | 5.2 | | |
| **Education Level** | 1524 | 82.8 | | 309 | 17.2 | | **0.001** |
| Primary | 386 | 25.3 | | 192 | 62.1 | | |
| Secondary | 695 | 45.6 | | 85 | 27.5 | | |
| Pre-university | 285 | 18.7 | | 18 | 5.8 | | |
| University | 158 | 10.4 | | 14 | 4.6 | | |

Continuous measured using independent t-test, categorical measured using chi-square. Bold values are statistically significant.

## HRQOL

The participants scored an overall mean EQ-5D index score of 0.93 (0.15) and a mean EQ-VAS score of 83.76 (12.77). The highest possible score of EQ-5D index score of 1 was reported at 68.1%, whereas the highest possible EQ-VAS score of 100 was at 16.1%. Age, gender, ethnicity, marital status, employment status, chronic diseases and education level were significantly associated with the EQ-5D index ($p<0.05$), whereas only the number of chronic diseases was seen to be significantly associated with EQ-VAS ($p = 0.001$) as seen in Table 2.

In Table 3, age ($p = 0.001$), gender ($p = 0.010$), marital status ($p = 0.001$), employment status ($p = 0.001$), chronic diseases ($p = 0.001$) and education level ($p = 0.001$) were significantly associated with problems reported for mobility. For self-care, age ($p = 0.001$), marital status ($p = 0.001$), employment status ($p = 0.001$) and chronic diseases ($p = 0.001$) were reported as problems. For usual activities, age ($p = 0.001$), marital status ($p = 0.002$), employment status ($p = 0.001$), chronic diseases ($p = 0.001$) and education level ($p = 0.006$) were reported as problems. Similarly, pain or discomfort reported problems was significant ($p = 0.001$) for age,

**Table 2. Characteristics of participants' EQ-5D index and Visual Analogue (VAS) score.**

| Independent Variables | EQ-5D Index | | Visual Analogue Scale (VAS) | |
|---|---|---|---|---|
| | Mean (SD) | *p*-value | Mean (SD) | *p*-value |
| **Total** | 0.93 (0.15) | | 83.76 (12.77) | |
| **Age** | | **0.001** | | 0.665 |
| Less than 60 years old | 0.95 (0.13) | | 84.00 (12.60) | |
| 60 years and older | 0.89 (0.16) | | 83.71 (12.62) | |
| **Gender** | | **0.006** | | 0.325 |
| Male | 0.94 (0.15) | | 83.62 (12.78) | |
| Female | 0.93 (0.13) | | 84.05 (12.76) | |
| **Ethnicity** | | **0.034** | | 0.223 |
| Malay / Bumiputera | 0.94 (0.14) | | 83.61 (12.82) | |
| Chinese | 0.89 (0.21) | | 85.22 (12.54) | |
| Indian | 0.94 (0.15) | | 83.83 (12.32) | |
| **Marital Status** | | **0.001** | | 0.789 |
| Married | 0.94 (0.15) | | 83.94 (12.60) | |
| Unmarried | 0.95 (0.13) | | 83.59 (13.21) | |
| Divorced / separated | 0.90 (0.16) | | 83.15 (13.13) | |
| **Employment** | | **0.001** | | 0.279 |
| Yes | 0.95 (0.14) | | 84.04 (12.54) | |
| No | 0.90 (0.17) | | 83.12 (13.30) | |
| **Chronic Disease** | | **0.001** | | **0.001** |
| 0 | 0.95 (0.14) | | 84.33 (12.58) | |
| 1 | 0.92 (0.14) | | 82.93 (12.92) | |
| 2 | 0.88 (0.17) | | 83.16 (12.73) | |
| 3 or more | 0.78 (0.27) | | 76.50 (14.58) | |
| **Education level** | | **0.001** | | 0.599 |
| Primary | 0.92 (0.17) | | 83.09 (13.726) | |
| Secondary | 0.94 (0.15) | | 83.91 (12.428) | |
| Pre-University | 0.95 (0.12) | | 84.14 (12.574) | |
| University | 0.96 (0.08) | | 85.30 (11.221) | |

Bold values are statistically significant (*p*-value < 0.05).

**Table 3. Percentage of reported problems among younger and older groups in EQ-5D.**

| | Mobility | | | Self-care | | | Usual Activities | | | Pain / Discomfort | | | Anxiety / Depression | | | Total Index Score | | |
|---|---|---|---|---|---|---|---|---|---|---|---|---|---|---|---|---|---|---|
| | Young | Older | *p*-value | Young | Older | *p*-value | Young | Older | *p*-value | Young | Older | *p*-value | Young | Older | *p*-value | Young | Older | *p*-value |
| **Total, (n)** | 9.7 (152) | 32.1 (105) | **0.001** | 3.8 (59) | 11.6 (38) | **0.001** | 9.1 (142) | 24.5 (80) | **0.001** | 16.5 (258) | 38.8 (127) | **0.001** | 13.5 (210) | 21.4 (70) | **0.001** | 27.0 (423) | 51.4 (168) | **0.001** |
| **Gender** | | | **0.010** | | | 0.063 | | | 0.078 | | | 0.083 | | | 0.117 | | | **0.035** |
| Male | 9.2 | 26.2 | | 3.4 | 9.5 | | 8.6 | 21.3 | | 15.6 | 36.7 | | 12.6 | 20.4 | | 24.9 | 49.3 | |
| Female | 10.7 | 44.3 | | 4.5 | 16.0 | | 10.0 | 31.1 | | 18.4 | 43.4 | | 15.2 | 23.6 | | 31.3 | 55.7 | |
| **Ethnicity** | | | 0.156 | | | 0.497 | | | 0.318 | | | 0.779 | | | 0.935 | | | 0.050 |
| Malay/Bumi | 9.6 | 33.6 | | 4.1 | 12.2 | | 9.3 | 26.1 | | 16.6 | 41.2 | | 13.5 | 22.3 | | 27.7 | 53.8 | |
| Chinese | 12.9 | 30.2 | | 3.0 | 9.4 | | 9.9 | 22.6 | | 15.8 | 34.0 | | 13.9 | 15.1 | | 26.7 | 49.1 | |
| Indian | 9.4 | 29.0 | | 1.6 | 9.7 | | 7.0 | 16.1 | | 15.6 | 32.3 | | 13.3 | 25.8 | | 20.3 | 38.7 | |
| **Marital Status** | | | **0.001** | | | **0.001** | | | **0.002** | | | **0.001** | | | **0.001** | | | **0.001** |
| Married | 9.5 | 28.6 | | 3.1 | 9.3 | | 8.3 | 21.6 | | 15.9 | 36.6 | | 12.6 | 19.4 | | 25.8 | 48.9 | |
| Unmarried | 5.0 | 12.5 | | 5.0 | 12.5 | | 10.7 | 25.0 | | 14.6 | 25.0 | | 13.1 | 0.0 | | 24.5 | 25.0 | |
| Divorced | 17.6 | 42.4 | | 5.9 | 17.4 | | 11.2 | 31.5 | | 22.5 | 45.7 | | 18.8 | 28.3 | | 37.4 | 59.8 | |
| **Employment** | | | **0.001** | | | **0.001** | | | **0.001** | | | **0.001** | | | 0.220 | | | **0.009** |
| Yes | 7.5 | 23.2 | | 2.3 | 7.4 | | 7.9 | 17.9 | | 15.6 | 31.6 | | 13.9 | 17.9 | | 25.3 | 46.3 | |
| No | 17.8 | 35.8 | | 9.4 | 13.4 | | 13.3 | 27.2 | | 19.9 | 41.8 | | 11.8 | 22.8 | | 33.2 | 53.4 | |
| **Chronic Disease** | | | **0.001** | | | **0.001** | | | **0.001** | | | **0.001** | | | **0.001** | | | **0.001** |
| 0 | 6.3 | 22.1 | | 3.3 | 6.2 | | 6.7 | 17.2 | | 12.6 | 26.2 | | 11.1 | 14.5 | | 21.7 | 38.6 | |
| 1 | 18.8 | 34.0 | | 4.4 | 15.1 | | 13.2 | 24.5 | | 26.4 | 42.5 | | 17.7 | 20.8 | | 41.6 | 55.7 | |
| 2 | 23.8 | 40.7 | | 6.6 | 11.9 | | 19.8 | 32.2 | | 31.4 | 52.5 | | 27.6 | 32.2 | | 46.2 | 64.4 | |
| 3 or more | 20.8 | 76.5 | | 8.3 | 35.3 | | 33.3 | 58.8 | | 41.7 | 76.5 | | 25.0 | 47.1 | | 50.0 | 88.2 | |
| **Education level** | | | **0.001** | | | 0.069 | | | **0.006** | | | **0.001** | | | **0.047** | | | **0.001** |
| Primary | 15.1 | 32.8 | | 4.1 | 11.5 | | 10.1 | 25.0 | | 20.7 | 38.5 | | 16.1 | 21.4 | | 33.4 | 53.1 | |
| Secondary | 8.5 | 32.1 | | 3.6 | 8.3 | | 7.8 | 19.0 | | 17.8 | 33.3 | | 12.8 | 15.5 | | 25.4 | 44.0 | |
| Pre-university | 6.7 | 33.3 | | 3.9 | 16.7 | | 10.5 | 33.3 | | 11.6 | 50.0 | | 10.5 | 44.4 | | 24.9 | 55.6 | |
| University | 6.3 | 14.3 | | 1.9 | 7.1 | | 8.9 | 21.4 | | 12.7 | 35.7 | | 17.1 | 14.3 | | 25.3 | 35.7 | |

Bold values are statistically significant (*p*-value < 0.05).

marital status, employment status, chronic diseases, and education level. For anxiety or depression reported problems for age (*p* = 0.001), marital status (*p* = 0.001), chronic diseases (*p* = 0.001) and education level (*p* = 0.047). Finally, total EQ-5D showed the problems reported for age (*p* = 0.001), gender (*p* = 0.035), marital status (*p* = 0.001), employment status (*p* = 0.009), number of chronic diseases (*p* = 0.001) and education (*p* = 0.001)

## Multivariate logistic regression analysis

Each dimension of the EQ-5D tool has been dichotomised as to having: (1) No problems at all and (2) Having some to severe problems. Independent variables of interest were age, gender, ethnicity, marital status, employment status, number of chronic diseases and education level. Only the final model of the backward stepwise regression and variables which exerted a significant relationship with any dimension from EQ-5D was reported in Table 4.

**Table 4. Multivariate logistic regression on the relationship of EQ-5D and influencing factors.**

| Influencing factors (final model) | | B | SE | *p*-value | Odds ratio | 95%CI |
|---|---|---|---|---|---|---|
| **Mobility** | | | | | | |
| Age | Less than 60 years old | 0 | | | 1 | |
| | More than 60 years old | 0.712 | 0.177 | **0.001** | 2.038 | 1.439–2.885 |
| Marital status | Married | 0 | | | 1 | |
| | Unmarried | - 0.562 | 0.296 | 0.058 | 0.570 | 0.319–1.018 |
| | Divorced / Separated | 0.493 | 0.179 | **0.006** | 1.637 | 1.152–2.325 |
| Employment status | Yes | 0 | | | 1 | |
| | No | 0.713 | 0.163 | **0.001** | 2.040 | 1.481–2.810 |
| Number of diseases | No disease | 0 | | | 1 | |
| | 1 disease | 0.983 | 0.174 | **0.001** | 2.673 | 1.902–3.756 |
| | 2 diseases | 1.249 | 0.215 | **0.001** | 3.488 | 2.290–5.312 |
| | 3 or more diseases | 1.677 | 0.357 | **0.001** | 5.350 | 2.659–10.763 |
| **Self-care** | | | | | | |
| Age | Less than 60 years old | 0 | | | 1 | |
| | More than 60 years old | 0.440 | 0.258 | 0.088 | 1.553 | 0.936–2.574 |
| Marital status | Married | 0 | | | 1 | |
| | Unmarried | 0.589 | 0.321 | 0.066 | 1.803 | 0.961–3.384 |
| | Divorced / Separated | 0.515 | 0.256 | **0.044** | 1.674 | 1.013–2.765 |
| Employment status | Yes | 0 | | | 1 | |
| | No | 1.199 | 0.244 | **0.001** | 3.318 | 2.055–5.375 |
| Number of diseases | No disease | 0 | | | 1 | |
| | 1 disease | 0.535 | 0.261 | **0.040** | 1.708 | 1.023–2.850 |
| | 2 diseases | 0.552 | 0.332 | 0.097 | 1.737 | 0.905–3.333 |
| | 3 or more diseases | 1.292 | 0.444 | **0.004** | 3.641 | 1.527–8.685 |
| **Usual Activities** | | | | | | |
| Age | Less than 60 years old | 0 | | | 1 | |
| | More than 60 years old | 0.672 | 0.189 | **0.001** | 1.957 | 1.353–2.832 |
| Employment status | Yes | 0 | | | 1 | |
| | No | 0.387 | 0.173 | **0.025** | 1.473 | 1.050–2.066 |
| Number of diseases | No disease | 0 | | | 1 | |
| | 1 disease | 0.674 | 0.185 | **0.001** | 1.962 | 1.364–2.821 |
| | 2 diseases | 1.076 | 0.223 | **0.001** | 2.933 | 1.895–4.539 |
| | 3 or more diseases | 1.856 | 0.348 | **0.001** | 6.400 | 3.236–12.659 |
| **Pain / Discomfort** | | | | | | |
| Age | Less than 60 years old | 0 | | | 1 | |
| | More than 60 years old | 0.807 | 0.144 | **0.001** | 2.241 | 1.690–2.972 |
| Number of diseases | No disease | 0 | | | 1 | |
| | 1 disease | 0.824 | 0.146 | **0.001** | 2.279 | 1.712–3.034 |
| | 2 diseases | 1.145 | 0.188 | **0.001** | 3.141 | 2.171–4.544 |
| | 3 or more diseases | 1.787 | 0.335 | **0.001** | 5.972 | 3.098–11.513 |
| **Anxiety / Depression** | | | | | | |
| Age | Less than 60 years old | 0 | | | 1 | |
| | More than 60 years old | 0.210 | 0.171 | 0.220 | 1.234 | 0.882–1.727 |
| Marital status | Married | 0 | | | 1 | |
| | Unmarried | 0.085 | 0.205 | 0.678 | 1.089 | 0.728–1.628 |
| | Divorced / Separated | 0.434 | 0.175 | **0.013** | 1.544 | 1.096–2.174 |

(*Continued*)

**Table 4.** (Continued)

| Influencing factors (final model) | | B | SE | *p*-value | Odds ratio | 95%CI |
|---|---|---|---|---|---|---|
| Number of diseases | No disease | 0 | | | 1 | |
| | 1 disease | 0.443 | 0.170 | **0.009** | 1.557 | 1.115–2.173 |
| | 2 diseases | 1.062 | 0.202 | **0.001** | 2.893 | 1.945–4.301 |
| | 3 or more diseases | 1.209 | 0.354 | **0.001** | 3.349 | 1.673–6.704 |
| **Total Index Score** | | | | | | |
| Age | Less than 60 years old | 0 | | | 1 | |
| | More than 60 years old | 0.523 | 0.154 | **0.001** | 1.687 | 1.248–2.282 |
| Gender | Male | 0 | | | 1 | |
| | Female | 0.230 | 0.114 | **0.043** | 1.259 | 1.008–1.574 |
| Ethnicity | Malay | 0 | | | 1 | |
| | Chinese | -0.22 | 0.192 | 0.910 | 0.672 | 0.672–1.425 |
| | Indian | -0.559 | 0.209 | **0.007** | 0.380 | 0.380–0.861 |
| Employment status | Yes | 0 | | | 1 | |
| | No | 0.224 | 0.129 | 0.083 | 1.251 | 0.972–1.610 |
| Number of diseases | No diseases | 0 | | | 1 | |
| | 1 disease | 0.863 | 0.133 | **0.001** | 2.382 | 1.836–3.089 |
| | 2 diseases | 1.059 | 0.180 | **0.001** | 2.885 | 2.027–4.106 |
| | 3 or more diseases | 1.566 | 0.350 | **0.001** | 4.787 | 2.413–9.497 |
| Education level | Primary | 0 | | | 1 | |
| | Secondary | -0.317 | 0.128 | **0.013** | 0.728 | 0.567–0.936 |
| | Pre-university | -0.240 | 0.169 | 0.155 | 0.787 | 0.565–1.095 |
| | University | -0.141 | 0.207 | 0.495 | 0.868 | 0.579–1.302 |

Bold values are statistically significant (*p*-value < 0.05).

The final model of the logistical regression analysis revealed an association between mobility and the older group (OR = 2.038, 95% CI:1.439–2.885), the divorced or separated (OR = 1.637, 95% CI:1.152–2.325), lack of employment (OR = 2.040, 95% CI:1.481–2.810) and having one (OR = 2.673, 95% CI:1.902–3.756), two (OR = 3.488, 95% CI:2.290–5.312), and three or more diseases (OR = 5.350, 95% CI:2.659–10.753). For self-care, older age was no longer significantly associated with reduced QOL, after adjustment for being divorced or separated (OR = 1.674, 95% CI:1.013–2.765), lack of employment (OR = 3.318, 95% CI:2.055–5.375) and having one (OR = 1.708, 95% CI:1.023–2.850) or three chronic diseases (OR = 3.641, 95% CI:1.527–8.685). For usual activities, being 60 years or older (OR = 1.957, 95% CI:1.353–2.832) remained significantly associated with reported problems in EQ-5D, after adjustment for lack of employment (OR = 1.473, 95% CI:1.050–2.066) and having one (OR = 1.962, 95% CI:1.364–2.821), two (OR = 2.933, 95% CI:1.895–4.539), and three or more diseases (OR = 6.400, 95% CI:3.236–12.659).

For pain or discomfort, being 60 years or older (OR = 2.241, 95% CI:1.690–2.972) and having one (OR = 2.279, 95% CI:1.712–3.034), two (OR = 3.141, 95% CI:2.171–4.544), and three or more diseases (OR = 5.972, 95% CI:3.098–11.513) were reported as problems. Within the anxiety/depression domain, being older was no longer significantly associated with reported problems in EQ5D, after adjustments for being divorced or separated (OR = 1.544, 95% CI:1.096–2.174) and having one (OR = 1.557, 95% CI:1.115–2.173), two (OR = 2.893, 95% CI:1.945–4.301), and three or more diseases (OR = 3.349, 95% CI:1.673–6.704). Finally, the total index score reported problems for those aged 60 years and older (OR = 1.687, 95% CI:1.248–2.282), female (OR = 1.259, 95% CI:1.008–1.574), have one (OR = 2.382, 95%

CI:1.836–3.089), two (OR = 2.885, 95% CI:2.027–4.106), and three or more diseases (OR = 4.787, 95% CI:2.413–9.497). Malays and those with primary level education reported higher figures when compared to Indians (OR = 0.380, 95% CI:0.380–0.861) and those with secondary school education (OR = 0.728, 95% CI:0.567–0.936).

## Discussion

Within the lower-income households surveyed in this representative, nationwide study, head of households aged 60 years and over were significantly more likely to report problems within the EQ-5D domains of mobility, usual activities, and pain/discomfort. Whereas the increased likelihood of reported problems in the self-care and anxiety/depression domains was accounted for by marital status, employment, and number of diseases. Adjacent to the HRQOL dimensions, previous total index scores conducted among similar sociodemographic reported lower HRQOL [18]. The scores were similar among nonprescribed pharmaceutical customers [15] and reported exact figures for the normal population [12]. These studies, however, had a smaller sample size and sampling bias, which reduced the generalisability with the present study [12, 18]. In addition, these studies utilised the United Kingdom EQ-5D questionnaire health states scoring system [15] compared to the Malaysian version used in the present study [13]. This social tariff might not be valid due to the differences in socioeconomic status between countries, which would be reflected in the health states [15]. In addition, people in different countries may refer to the levels of health differently due to cultural differences [33–35].

Juxtaposed with the same study [18], age was not significantly associated with HRQOL. The counterevidence point towards the overwhelming effect of age on HRQOL [15, 20, 36–38], similar to the present study. The implications of an ageing phenomenon include structural changes to families and households, social networks and interaction, leisure, housing and transportation, welfare services and pension, saving and consumption behaviour, and labour markets [39]. Hence, it is imperative to uplift the QOL of the older generation. Interestingly, older age was no longer significant for self-care and anxiety/depression after adjusting for the covariates. Similar findings were found in China [20] and Switzerland [40]. As life expectancy has increased considerably, many challenges emerge, highlighting the need to organise healthcare and anticipate age-related health problems [20]. Therefore, older people receive better care, suffer from fewer disabilities or cope better with the limitations. There have been improvements in assistive technology for self-care, housing standards, transportation, as well as inclusive social policies [41]. The challenge is not living longer, but preserving the highest levels of QOL as long as possible [42, 43].

Significant associations between QOL and chronic diseases were also found in the present study. The presence of chronic disease is a contributing factor in many studies [18–20, 36]. The NHMS reported NCDs in 2015 and 2019, where 30% of Malaysians were hypertensive, 18.3% were diabetic, and 38.1% had high cholesterol levels [25, 44]. The presence of chronic diseases significantly affected the health and QOL caused by disease complications, limitation and acute problems [45–47]. As chronic diseases are strongly linked with physical health and QOL [24], this increasing trend is more detrimental to the lower-income group [25]. This is because the lower-income group is more vulnerable to financial hardships as their income is sufficient for basic needs but not protecting them from risky scenarios such as unforeseen medical costs and other out-of-pocket expenditures [48].

Besides that, the odds of having problems with mobility, self-care and usual activities is significantly associated with unemployment. Although the total HRQOL index score was not significant when adjusted with other factors, the unemployed exhibit lower HRQOL similar to

other studies that adjust for gender and socioeconomic status [2, 45, 49]. Unemployment leads to lower economic output causing stress and unhappiness, which leads to lower QOL [50]. Adjacent to unemployment findings, studies have found that lower educational levels reported more HRQOL problems [26, 32, 36] while in others, no association was found [18, 20]. Studies showed that higher education is associated with better health status among older individuals [37, 51]. In our study, findings showed some association, particularly when compared between primary and secondary level education. Therefore, approaches such as social, health care, and legal systems are imperative to protect the health and welfare of these groups [52].

A strong sampling method and large sample size allow a strong representation of HRQOL among low-income households. Conversely, a low response from other ethnic groups based on the initial DOSM sampling frame reduces generalisability among the entire ethnically diverse Malaysian population and therefore should be interpreted with caution, particularly among the non-Malays. In addition, the use of EQ-5D is associated with shortcomings such as having a larger ceiling effect. This reflects the conceptualisation of the EQ-5D instrument which focuses on limitations in function and symptoms and does not include positive aspects of health such as wellbeing and motivation or energy to work or engage with activities [53]. Besides that, self-reporting of medical illnesses during the survey were not fully defined or medically diagnosed. Finally, the duration of the data collection period varied, and thus changes in the QOL may have dropped off with the extension of time due to factors such as rising COVID-19 cases which led to reduced economic output.

Nonetheless, the results of this study have provided indicative information to policymakers and researchers alike of HRQOL profile among low-income Malaysians. Lower income older individuals particularly among females, with lower education and with co-morbidities exhibit lower overall HRQOL and thus should be prioritised moving forward. Local initiatives such as PeKa B40, mySalam and household living aid (BSH) should be continuously improved and disseminate among the disadvantaged group. By providing opportunities to age actively through financial aid, voluntary activities, community engagement, and improving health and financial literacies, policymakers and health practitioners can mobilise effort to reduce health inequity among all Malaysians.

## Conclusion

Older individuals within lower-income households exhibit lower HRQOL compared to younger people. Therefore, enhancing QOL of lower-income older communities requires the combined effort of individual and policy-making levels. The approach to elevating individual QOL can be achieved through voluntary activities, community engagement, and intervention programmes to include disengaged individuals in local activities. Integrating health promotion and financial literacy in these activities is essential for expanding health and financial security and funnelling funds more efficiently to improve QOL. The policymakers should utilise a comprehensive approach via the active ageing concept at all levels to improve the lives of many disadvantaged Malaysians.

## Supporting information

**S1 Table. EQ-5D questionnaire in Malay.**
(DOCX)

**S2 Table. Characteristics of study sample.**
(DOCX)

**S1 Appendix. Dataset.**
(XLSX)

## Acknowledgments

The authors thank all the participants that participated in the survey and enumerators that engage the community and collected the data.

## Author Contributions

**Conceptualization:** Hussein Rizal, Mas Ayu Said, Hazreen Abdul Majid, Tin Tin Su, Tan Maw Pin, Rozmi Ismail.

**Data curation:** Hussein Rizal, Mas Ayu Said, Hazreen Abdul Majid.

**Formal analysis:** Hussein Rizal.

**Funding acquisition:** Mas Ayu Said, Tin Tin Su, Mohd Azlan Shah Zaidi.

**Investigation:** Hussein Rizal, Mas Ayu Said, Tan Maw Pin.

**Methodology:** Hussein Rizal, Mas Ayu Said, Hazreen Abdul Majid, Tin Tin Su, Tan Maw Pin.

**Project administration:** Mas Ayu Said, Hazreen Abdul Majid, Mohd Azlan Shah Zaidi.

**Validation:** Tan Maw Pin.

**Visualization:** Tan Maw Pin.

**Writing – original draft:** Hussein Rizal, Mas Ayu Said, Hazreen Abdul Majid, Tin Tin Su, Tan Maw Pin, Rozmi Ismail, Mohd Azlan Shah Zaidi.

**Writing – review & editing:** Hussein Rizal, Mas Ayu Said, Hazreen Abdul Majid, Tin Tin Su, Tan Maw Pin, Rozmi Ismail, Mohd Azlan Shah Zaidi.

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
