## [Decision Letter · Decision Letter 0]

23 Dec 2021

PONE-D-21-32914Health-Related Quality of Life of Younger and Older Lower-Income Households in MalaysiaPLOS ONE

Dear Dr. Said,

Thank you for submitting your manuscript to PLOS ONE. After careful consideration, we feel that it has merit but does not fully meet PLOS ONE’s publication criteria as it currently stands. Therefore, we invite you to submit a revised version of the manuscript that addresses the points raised during the review process.

ACADEMIC EDITOR:The reviewers have raised some valid comments to this manuscript. Authors are invited to carefully address these comments and revise the manuscript accordingly before resubmission.

We look forward to receiving your revised manuscript.

Kind regards,

Hoh Boon-Peng, PhD

Academic Editor

PLOS ONE

“This research was funded by the Long-Term Research Grant (LRGS) awarded to the Malaysia Research University Network (MRUN) approved by the Ministry of Education, Malaysia, Grant Number: LGRS MRUN/F1/01/2019.”

“Mas Ayu Said (corresponding author) received the Long Term Research Grant Scheme (LR004-2019). The funder played no role in the study design, data collection and analysis, decision to publish, or preparation of the manuscript.

https://mastic.mosti.gov.my/sti-incentive/long-term-research-grant-scheme-lrgs”

Reviewers' comments:

Reviewer's Responses to Questions

**Comments to the Author**

1. Is the manuscript technically sound, and do the data support the conclusions?

Reviewer #1: Yes

Reviewer #2: Partly

2. Has the statistical analysis been performed appropriately and rigorously? 

Reviewer #1: Yes

Reviewer #2: Yes

3. Have the authors made all data underlying the findings in their manuscript fully available?

Reviewer #1: Yes

Reviewer #2: Yes

4. Is the manuscript presented in an intelligible fashion and written in standard English?

Reviewer #1: Yes

Reviewer #2: Yes

5. Review Comments to the Author

Reviewer #1: The article Health-Related Quality of Life of Younger and Older Lower-Income Households in Malaysia is worth publishing, with the following suggestions for clarification, which may stand somewhere between minor and major changes:

In 88 – 90 authors properly stated that: Empirical analyses concerning HRQOL could further extend the understanding of health inequalities. They indicate that low health status groups are faced with a double burden, first by increased levels of health impairments and second by lower levels of HRQOL once health is impaired.

I think here a bit more should be elaborated as this is important for the conclusion too. The WHO and European Commission define healthcare inequalities as the, “differences in health status or in the distribution of health determinants between different population groups. When avoidable, these inequalities are termed health inequities.”

Health inequalities are ultimately about differences in the status of people’s health. Health inequalities are neither inevitable and any gaps are not fixed. Evidence shows that a comprehensive approach to tackling them can make a difference. / https://www.kingsfund.org.uk/publications/what are health inequalities/

The causes of health inequalities are often multi factorial and can be both secondary to individual characteristics and external factors (determinants).

The determinants of health inequalities can be broadly categorized as health related and wider (non health/social) determinants of health (see proposed references):

Bulletin of the World Health Organization2018;96:654-659. doi:http://dx.doi.org/10.2471/BLT.18.210401

WHO Health Inequality and Inequity World Health Organization, Health Impact Assessment Glossary, Health Inequality and inequity

NHS England definition of inequalities https://www.england.nhs.uk/ltphimenu/definitions-for-health-inequalities/

Addressing health inequalities in the European Union: concepts, action, state of play. European Parliament (2020) ISBN: 978-92-846-6343-9

Social determinants of health: Key concepts World Health Organization (2013) https://www.who.int/news-room/q-a-detail/social-determinants-of-health-key-concepts

Health Topics: Social determinants of health WHO https://www.who.int/health-topics/social-determinants-of-health#tab=tab_1

Report of the Working Group on Inequalities in Health,Department of Health and Social Security (UK), 1980 (The ‘Black Report’)

Grove J, ClaesonM, Bryce J, AmouzouA, BoermaT, WaiswaP, et al.; Kirkland Group. Maternal, newborn, and child health and the Sustainable Development Goals –a call for sustained and improved measurement. Lancet. 2015 Oct 17;386(10003):1511–4.http://dx.doi.org/10.1016/S0140-6736(15)005176pmid:26530604

Australia’s health 2018: in brief Australian Institute of Health and Welfare (2018) ISBN:978-1-76054-377-8

More generally, I would acknowledge authors to point out the limitations of the method. EQ-5D is associated with shortcomings, let alone the dichotomization.

I would recommend in a further analysis the use of estimands to embed and adequately accounts for the intercurrent events.

In 159 – typo, approved by the Universiti Malaya Research

Reviewer #2: 1. Abstract (lines 31-32): Please add the age range of the sample in the Methods sub-section

2. Introduction (line 53): Please provide the full term of QOL in its first appearance in the Main Manuscript.

3. Introduction: The explanation about ageing population and health inequalities in Malaysia and what is the importance of looking at quality of life among older people in the country is missing. It is important to understand the context of the study place in this study.

4. Methods: Please provide the age range of the sample. The term “aged less than 60 years old” could be incorrect here as it means that the age group included those age 0-59 years. The age range of the sample thus required to identify the correct age group.

5. Methods: As the sample is designed as a purposive random sampling, what are the reasons of significant difference in the composition of ethnicity between the older and younger group (Table 1)? How will it affect the findings?

6. Methods: The marital status was defined into (1) married; (2) unmarried; and (3) divorced/separated? How about the respondents who are widowed? Where are they in this classification?

7. Methods: There is missing values in the data. For example, the data on education level among younger age group is only available for 1524 from 1571 respondents. The same data is only available for 309 from the total 328 respondents in the older age group. All the missing values should be reported and explanation on how those missing values addressed in this study should be available. The ignorance of missing values will lead to the biases in the analysis.

8. Results: The aim of this study is to identify the difference in QOL between two different age groups. The logistic regression thus could be performed separately in those two different age groups to follow up the results from Tables 2 and 3.

9. Discussion: What are the implications of these findings for policy-makers and public health practitioners in Malaysia in addition of as the “indicative information” (lines 293-294)? How will it affect the health and well being given the ageing population in the country? How big is the B40 group in the country that may affected?

6. PLOS authors have the option to publish the peer review history of their article (what does this mean?). If published, this will include your full peer review and any attached files.

Reviewer #1: **Yes: **Borislav Borissov

Reviewer #2: No

---

## [Author Response · Author response to Decision Letter 0]

11 Jan 2022

Reviewer 1 Comments Changes Revision/Explanation

1. The article Health-Related Quality of Life of Younger and Older Lower-Income Households in Malaysia is worth publishing, with the following suggestions for clarification, which may stand somewhere between minor and major changes:

In 88 – 90 authors properly stated that: Empirical analyses concerning HRQOL could further extend the understanding of health inequalities. They indicate that low health status groups are faced with a double burden, first by increased levels of health impairments and second by lower levels of HRQOL once health is impaired. I think here a bit more should be elaborated as this is important for the conclusion too. Pg 4, line 90-98

 Thank you for your comments. We have elaborated on the point on health inequalities by stating the following:

Added “The interactions between differing kinds of inequality and its factors are often complex, multidirectional, and inter-related. People can find it more difficult to move away from unhealthy behaviours if they are worse off in a wider range of determinants of health. For example, lower socio-economic groups associated to have a higher prevalence of risky health behaviours, poor diet, limited access to care and less opportunity to lead healthy lives. In addition, disadvantages are concentrated in particular sectors of the population and can be mutually reinforcing [28].” This means that the low-income population living in close proximity together reinforces their accumulative disadvantage such as being in their psychological comfort zone, lower education quality in terms of access to better schools and to afford tuitions, and limited occupational opportunities due to lower education and skill level.

Source: https://www.kingsfund.org.uk /publications/what-are-health-inequalities

2. More generally, I would acknowledge authors to point out the limitations of the method. EQ-5D is associated with shortcomings, let alone the dichotomization. Pg 15, line 326-330

 Added: “In addition, the use of EQ-5D is associated with shortcomings such as having a larger ceiling effect. This reflects the conceptualisation of the EQ-5D instrument which focuses on limitations in function and symptoms and does not include positive aspects of health such as wellbeing and motivation or energy to work or engage with activities [53].”

3. I would recommend in a further analysis the use of estimands to embed and adequately accounts for the intercurrent events.

 No changes

 The authors recognise the value of the suggested analysis but have decided not to expand on the already extensive analysis conducted in this manuscript. We may look further opportunity to write the separate manuscript based on the reviewer’s suggestion

4. In 159 – typo, approved by the Universiti Malaya Research Pg 7, line 179 Amended: Universiti Malaya is the Malay name of the University. Now we change it to the English name “University of Malaya”

Reviewer 2 Comments Changes Revision/Explanation

1. Abstract (lines 31-32): Please add the age range of the sample in the Methods sub-section Pg 2, line 30

 Added: “aged 18 years and older”

2. Introduction (line 53): Please provide the full term of QOL in its first appearance in the Main Manuscript. Pg 3, line 50

 Added: “quality of life’

3. Introduction: The explanation about ageing population and health inequalities in Malaysia and what is the importance of looking at quality of life among older people in the country is missing. It is important to understand the context of the study place in this study.

 Pg 2, line 51-54 and Pg 4, line 90-98

 Added: “Older people have higher probability of suffering from multiple health disorders due to experiencing reduced physical and mental functions. Loneliness, impaired sexual activity and chronic metabolic disorders are some of the causes that can result in emotional disturbances which lead to reduced QOL [3].”

Added: The interactions between differing kinds of inequality and its factors are often complex, multidirectional, and inter-related. People can find it more difficult to move away from unhealthy behaviours if they are worse off in a wider range of determinants of health. For example, lower socio-economic groups associated to have a higher prevalence of risky health behaviours, poor diet, limited access to care and less opportunity to lead healthy lives. In addition, disadvantages are concentrated in particular sectors of the population and can be mutually reinforcing [28]. This means that the low-income population living in close proximity together reinforces their accumulative disadvantage such as being in their psychological comfort zone, lower education quality in terms of access to better schools and to afford tuitions, and limited occupational opportunities due to lower education and skill level.

4. Methods: Please provide the age range of the sample. The term “aged less than 60 years old” could be incorrect here as it means that the age group included those age 0-59 years. The age range of the sample thus required to identify the correct age group. Pg 5, line 108; and Pg 6, line 155

 Added: “aged 18 years and older” and added: “aged 18 to 59 years old”

5. Methods: As the sample is designed as a purposive random sampling, what are the reasons of significant difference in the composition of ethnicity between the older and younger group (Table 1)? How will it affect the findings? Pg 15, line 324-326 

 Explained and expanded reasoning: “A low response from other ethnic groups based on the initial DOSM sampling frame reduces generalisability among the entire ethnically diverse Malaysian population and therefore should be interpreted with caution.”

6. Methods: The marital status was defined into (1) married; (2) unmarried; and (3) divorced/separated? How about the respondents who are widowed? Where are they in this classification? No changes

 The authors have included the term widowed as part of the definition of separated and thus is included in the (3) divorced/separated category.

7. Methods: There is missing values in the data. For example, the data on education level among younger age group is only available for 1524 from 1571 respondents. The same data is only available for 309 from the total 328 respondents in the older age group. All the missing values should be reported and explanation on how those missing values addressed in this study should be available. The ignorance of missing values will lead to the biases in the analysis. Pg 6, line 150-151 and Pg 8, line 196-198 (Table 1)

 Added: “Frequency analysis was carried out to identify missing values and extreme values. Extreme values that were deemed as a case of wrong data entry were deleted and assigned a missing value.” The full dataset will be provided as a supplementary to provide transparency.

Added: Total (percentage), n (%) for each variable description to accurately describe missing data for each variable.

8. Results: The aim of this study is to identify the difference in QOL between two different age groups. The logistic regression thus could be performed separately in those two different age groups to follow up the results from Tables 2 and 3. No changes Originally, the manuscript included the logistic regression (LR) for both age groups to determine their differences as stated in the aims of the study. However, conducting both young and old age groups independently would require a univariate chi-square analysis of each variable to determine its inclusion in the LR analysis, making the table 6 pages long. 

In addition, the independent LR analysis would make the analysis too broad in scope to discuss in-depth.

After deliberation, the authors decided to be more concise and articulate with the writeup for LR, focusing more on the relationship of each QOL dimension in regards with economic status of both age group.

9. Discussion: What are the implications of these findings for policy-makers and public health practitioners in Malaysia in addition of as the “indicative information” (lines 293-294)? How will it affect the health and well-being given the ageing population in the country? How big is the B40 group in the country that may affected? Pg 15-16, line 335-341 Added: “Lower income older individuals particularly among females, with lower education and with co-morbidities exhibit lower overall HRQOL and thus should be prioritised moving forward. Local initiatives such as PeKa B40, mySalam and household living aid (BSH) should be continuously improved and disseminate among the disadvantaged group. By providing opportunities to age actively through financial aid, voluntary activities, community engagement, and improving health and financial literacies, policymakers and health practitioners can mobilise effort to reduce health inequity among all Malaysians.”

---

## [Decision Letter · Decision Letter 1]

26 Jan 2022

Health-related quality of life of younger and older lower-income households in Malaysia

PONE-D-21-32914R1

Dear Dr. Said,

We’re pleased to inform you that your manuscript has been judged scientifically suitable for publication and will be formally accepted for publication once it meets all outstanding technical requirements.

Kind regards,

Hoh Boon-Peng, PhD

Academic Editor

PLOS ONE

Additional Editor Comments (optional):

Reviewers' comments:

Reviewer's Responses to Questions

**Comments to the Author**

1. If the authors have adequately addressed your comments raised in a previous round of review and you feel that this manuscript is now acceptable for publication, you may indicate that here to bypass the “Comments to the Author” section, enter your conflict of interest statement in the “Confidential to Editor” section, and submit your "Accept" recommendation.

Reviewer #1: (No Response)

Reviewer #2: All comments have been addressed

2. Is the manuscript technically sound, and do the data support the conclusions?

Reviewer #1: (No Response)

Reviewer #2: Yes

3. Has the statistical analysis been performed appropriately and rigorously? 

Reviewer #1: (No Response)

Reviewer #2: Yes

4. Have the authors made all data underlying the findings in their manuscript fully available?

Reviewer #1: (No Response)

Reviewer #2: Yes

5. Is the manuscript presented in an intelligible fashion and written in standard English?

Reviewer #1: (No Response)

Reviewer #2: Yes

6. Review Comments to the Author

Reviewer #1: (No Response)

Reviewer #2: Thank you for the revised manuscript and analysis. The author(s) has well responded to each of the comment.

7. PLOS authors have the option to publish the peer review history of their article (what does this mean?). If published, this will include your full peer review and any attached files.

Reviewer #1: **Yes: **Borislav Borissov

Reviewer #2: No

---

## [Editor Report · Acceptance letter]

31 Jan 2022

PONE-D-21-32914R1 

Health-related quality of life of younger and older lower-income households in Malaysia 

Dear Dr. Said:

I'm pleased to inform you that your manuscript has been deemed suitable for publication in PLOS ONE. Congratulations! Your manuscript is now with our production department. 

Kind regards, 

on behalf of

Dr. Hoh Boon-Peng 

Academic Editor

PLOS ONE